# Unveiling the Significance of HLA and KIR Diversity in Underrepresented Populations

**DOI:** 10.3390/biomedicines12061333

**Published:** 2024-06-15

**Authors:** Lucía Santiago-Lamelas, Patricia Castro-Santos, Ángel Carracedo, Jordi Olloquequi, Roberto Díaz-Peña

**Affiliations:** 1Fundación Pública Galega de Medicina Xenómica, SERGAS, Grupo de Medicina Xenomica-USC, Instituto de Investigación Sanitaria de Santiago (IDIS), 15706 Santiago de Compostela, Spain; luciasantiagolamelas1@gmail.com (L.S.-L.); patricassan@gmail.com (P.C.-S.); angel.carracedo@usc.es (Á.C.); 2Facultad de Ciencias de la Salud, Universidad Autónoma de Chile, Talca 3460000, Chile; 3Grupo de Medicina Xenómica, CIMUS, Universidade de Santiago de Compostela, 15782 Santiago de Compostela, Spain; 4Centre for Biomedical Network Research on Rare Diseases (CIBERER), Instituto de Salud Carlos III, 28029 Madrid, Spain; 5Departament de Bioquímica i Fisiologia, Facultat de Farmàcia i Ciències de l’Alimentació, Universitat de Barcelona, 08028 Barcelona, Spain

**Keywords:** underrepresented populations, HLA, KIR, genome-wide association studies, imputation

## Abstract

Human leukocyte antigen (HLA) molecules and their relationships with natural killer (NK) cells, specifically through their interaction with killer-cell immunoglobulin-like receptors (KIRs), exhibit robust associations with the outcomes of diverse diseases. Moreover, genetic variations in HLA and KIR immune system genes offer limitless depths of complexity. In recent years, a surge of high-powered genome-wide association studies (GWASs) utilizing single nucleotide polymorphism (SNP) arrays has occurred, significantly advancing our understanding of disease pathogenesis. Additionally, advances in HLA reference panels have enabled higher resolution and more reliable imputation, allowing for finer-grained evaluation of the association between sequence variations and disease risk. However, it is essential to note that the majority of these GWASs have focused primarily on populations of Caucasian and Asian origins, neglecting underrepresented populations in Latin America and Africa. This omission not only leads to disparities in health care access but also restricts our knowledge of novel genetic variants involved in disease pathogenesis within these overlooked populations. Since the KIR and HLA haplotypes prevalent in each population are clearly modelled by the specific environment, the aim of this review is to encourage studies investigating HLA/KIR involvement in infection and autoimmune diseases, reproduction, and transplantation in underrepresented populations.

## 1. Introduction

The immune system is formed by a high diversity of cells characterized by their capacity for interaction. Specifically, natural killer cells (NKs) contribute to both innate and adaptive immune responses. The NK cell response, in which cytokines and cytotoxic granules are released, is the result of binding of ligands to NK receptors. The relevant NK receptors are KIR, CD94/NKG2A, and leukocyte immunoglobulin-like receptor (LILRB1), among others. The main ligands of these receptors are human leukocyte antigen (HLA) molecules, the most relevant molecules involved in the adaptive immune response, which are characterized by high variability.

As a relevant part of the innate immune system, the NK response is characterized by as a very rapid first response to a pathogen. However, NK cells are also clearly involved in the adaptive response through interactions with HLA molecules. KIR binding to HLA expressed by a T-cell mostly triggers an inhibition of the NK cell response and activation of the T-cell (recognition of self). Conversely, low HLA expression in a cell caused by herpesvirus, retrovirus, or tumour cells causes release of NK granules (missing self) [1]. Additionally, expression of CD16 and the FcgammaRIII receptor by NKs allows these cells to participate in the antibody-dependent cellular cytotoxicity (ADCC) response [2].

For this reason, the HLA/KIR interaction is likely relevant for controlling the immune response and is clearly influenced by the high variability present in both pathways. In fact, historically, HLA alleles and, more recently, KIR alleles have been associated with susceptibility to or evolution of different diseases. The aim of this review is to summarize the current findings concerning the implications of HLA/KIR genes in the pathogenesis of different conditions. In particular, this review will focus on the gap between the knowledge of the role of these molecules in disease in Caucasian and Asian populations and underrepresented populations, such as Latin Americans or Africans.

## 2. Major Histocompatibility Complex

### 2.1. Structure and Function of HLA Molecules

The human major histocompatibility complex (MHC) region is located on the short arm of chromosome 6, containing a group of polymorphic genes that encodes core components of human system, including the HLA genes [3]. The HLA region is commonly subdivided into three subclasses of genes (classes I, II, and III). The HLA class I and II genes encode proteins involved in the presentation of antigenic peptides from invading pathogens to T cells, triggering the adaptive immune response, and also in the self-antigens presentation during immune cell education and selection. Therefore, HLA molecules are implicated in discrimination between self and nonself. Given this, HLA molecules are clearly involved in many processes, such as the response to pathogens and the etiopathogenesis of the response to infection, rejection (due to histocompatibility issues during organ transplant), autoimmune diseases, or indeed processes such as implantation or placentation in reproduction [4] (Figure 1).

In addition to being one of the most complex systems in the human genome, some of the genes in the MHC are the most polymorphic. Indeed, more than 38,000 different HLA class I and II alleles have been reported to date (http://hla.alleles.org/nomenclature/stats.html; last accessed on 2 February 2024). Most of these polymorphisms involve nonsynonymous amino acid changes in the peptide-binding groove of HLA molecules, showing strong selection pressure in this region and revealing the relevance of the HLA–peptide interaction.

The repertoire of peptides that HLA-class I molecules can bind to is known as the immunopeptidome, and its diversity determines the capacity of the immune system to respond to microorganisms and the implication of HLA molecules in autoimmunity, rejection, or fertility. Characteristically, the capacity of a peptide to be bound by a determinate HLA molecule is shaped by the amino acid residues that form pockets (A to F) in the HLA-class I peptide-binding groove (PBG) [5]. In particular, the B, C, and F pockets are restricted, while residues outside these pockets allow for increased immunopeptidome diversification [6]. Therefore, HLA-class I polymorphisms, especially polymorphisms in the B, C, and F pockets, determine the immunopeptidome and the immune response.

### 2.2. HLA Diversity and Disease Susceptibility

HLA genes show the highest variability in the human genome. Historically, diverse alleles of the HLA genes have been associated with disease susceptibility, mainly with autoimmunity (Figure 1). For instance, *HLA-B*27* is associated with ankylosing spondylitis (AS) [7], while *HLA-C*06:02* is linked to psoriasis and psoriatic arthritis (PsA) [8]. Similarly, *HLA-DRB1* alleles are associated with numerous autoimmune diseases, including rheumatoid arthritis (RA) [9,10], systemic lupus erythematosus (SLE) [11], and multiple sclerosis [12]. Additionally, specific HLA alleles contribute to the risk of celiac disease (*HLA-DQA1*, *-DQB1*) [13], narcolepsy (*HLA-DQB1*) [14], leprosy (*HLA-DRB1*10:01~HLA-DQA1*01:05* and -*DRB1*15:01*) [15], type 1 diabetes (*HLA-DRB1*, *-DQA1* and *-DQB1*) [16], Graves’ disease (*HLA-DRB1* and -*DQB1*) [17], and myasthenia gravis (*HLA-C*07:01* and -*DQB1*) [18]. Notably, *HLA-B*57:01* is linked to abacavir drug hypersensitivity [19].

The association between *HLA-B*27* and AS is the strongest known between an HLA class I molecule and a disease. In fact, more than 90% of Caucasians with AS are HLA-B27 positive [20]. The prevalence of *HLA-B27* varies between populations, being approximately 8% in white Europeans, highly variable in American populations (high among native tribes of North America but low in unmixed native tribes in South America) and rare in Africans and Australian Aboriginals [21,22]. The most widespread of the *HLA-B*27* subtypes is *HLA-B*27:05*, which is thought to be the ancestral subtype and is present in nearly all populations of the world. Most of the relatively common subtypes (*B*27:02*, *B*27:04*, *B*27:05* and *B*27:07*) have been associated with AS, though two subtypes, *B*27:06* (common in Asian southeast populations) and *B*27:09* (described only in Sardinia and southern Italy), are considered exceptions [7]. In the Asian population, the HLA-B27 incidence can be divided according to proximity to Europe. Thus, in West Russian regions (the closest to Europe), *HLA-B*27:05* is the most common HLA-B27 subtype; in the East Russian region, the strongest influence comes from Southeast Asia, as the most common subtype is *HLA-B*27:04* [23]. In general, the incidence of AS is proportional to the frequency of *HLA-B27* (Figure 2). Indeed, the low occurrence of *HLA-B27* in central and southern Africa (less than 1%) has traditionally been considered a reason for the rarity of AS throughout sub-Saharan Africa, but other *HLA-B* non-B*27 alleles have been associated with these populations [24,25]. Further studies in diverse ethnic populations, including a large number of AS patients and healthy controls, are crucial to comprehensively understand the role of *HLA-B*27* allotypes and non-*HLA-B*27* alleles in AS pathogenesis. This reinforces the necessity to broaden our genetic understanding of complex diseases in underrepresented populations.

The large genetic variation in HLA alleles is probably caused mainly by the adaptation of human populations to pathogen-rich environments [26]. In particular, some studies suggest that the response to many viral infections is driven by an association between HLA class I and CD8+ T cells [4]. In the case of HIV/AIDS alleles, *HLA-B*57:02*, *HLA-B*57:03,* and *HLA-B*58:01* have been described as protective against progression of HIV-related complications [26]. Interestingly, some alleles are protective in some populations but not in others. This may be due to viral escape mechanisms in which mutations appear in some populations and prevent specific HLA allele-associated protection [27]. In the case of hepatitis B virus (HBV) infection in European and African American individuals, the rs9277534 SNP, located in the 3′ untranslated region of the HLA-DPB1 gene, is associated with recovery from HBV infection when it is in linkage disequilibrium with *HLA-DPB1*01:01* but is related to the persistence of HBV with *HLA-DPB1*04:01* [28]. Another example is hepatitis C virus (HCV) infection, in which *HLA-DRB1*03:01* in combination with rs4273729, located in an untranslated region of HLA class II genes, was shown to be associated with persistence of the disease in African and European populations [26]. Also, an association between the HLA-B*1502 allele and carbamazepine-induced Stevens–Johnson syndrome and toxic epidermal necrolysis in Han-Chinese, Thai, and Malaysian populations [29]. The coevolution of HLA alleles and pathogens has been identified as one of the main reasons for HLA variability. Given the crucial role of HLA molecules in antigen presentation, it is likely that there are many associations between HLA alleles and the response to infection.

### 2.3. Genome-Wide Association Studies and Single-Nucleotide Polymorphisms

The pathogenesis of the great majority of complex diseases mentioned herein has a polygenic basis. In the last few years, our understanding of genetic susceptibility to these pathologies has been greatly improved by findings derived from powered genome-wide association studies (GWASs), which are based on single-nucleotide polymorphism (SNP) arrays. These studies have driven the discovery of many susceptibility loci, suggesting the possibility of identifying individuals at high risk for certain diseases [30]. One of the known findings of GWASs is that genetic variation within the HLA region plays an important role in mediating susceptibility to infectious and autoimmune diseases [4,31], as has been described for AS [20], psoriasis [8], SLE [32], multiple sclerosis [33], RA [34], and the response to infection [26]. In addition, the large volume of GWAS information has provided an opportunity to apply imputation tools, that is, to collect SNP data and subsequently infer HLA alleles. HLA imputation allowed us to predict HLA genotypes from genotyped GWAS SNPs through statistical inference. However, due to the high prevalence of HLA polymorphisms and the high variation across populations, HLA allele imputation requires a large amount of information from each population. It relies on reference panels of known SNPs, HLA haplotypes, and genotypes that allow for linking SNPs and HLA alleles, which are specific to each population; obviously, this approach is not suitable in many cases, especially in underrepresented populations [35]. The vast majority of GWASs have been conducted in European populations [36,37], and imputations increases in complexity when the population has high admixture, as is the case for Latin Americans. Additionally, most SNPs used in GWASs are common variants (defined as SNPs with minor allele frequencies > 0.05). This is exacerbated when underrepresented populations are surveyed because most arrays commonly provide poor coverage in non-Caucasian populations, and genotype imputation cannot be performed [38]. Consequently, the extraordinary diversity of HLA molecules and their crucial role in the immune response and disease development highlight the need for a deeper understanding of HLA allele distribution and its implications for the etiopathology of associated diseases in diverse populations. Achieving this objective will require continued efforts in several areas, including sequencing data collection from underrepresented populations, and the refinement of computational tools for accurate HLA imputation. Such comprehensive approaches are essential to bridge knowledge gaps and ensure equitable advancements in our understanding of HLA-associated diseases across all populations.

## 3. Killer Cell Immunoglobulin-like Receptors

### 3.1. KIR Structure

KIR genes are encoded on the long arm of chromosome 19 (19q13) and produce transmembrane receptors expressed on NK and some T cells. Specifically, KIR genes are located in the leukocyte receptor complex (LRC) and constitute a multigene family that achieves genomic diversity through variations in gene content and allelic polymorphism (Figure 3).

### 3.2. Haplotypic and Allelic Diversity

The region where the KIR genes are located is one of the most variable regions in the genome, both in terms of the number of genes and allelic polymorphisms. Fourteen functional KIR genes (some of which had an elevated number of alleles) and two pseudogenes have been described, but not all KIR genes are present in each individual, increasing the number of different genotypes [39]. In this way, each NK cell clone can express several different KIR receptors depending on its gene content and polymorphism [40]. This finding implies the potential generation of a wide repertoire of haplotypes and NK cell clones in which KIRs are expressed at the cell surface in a combinatorial mode. In fact, the functional heterogeneity of NKs regarding their KIRs includes variable expression, ligand-binding specificity determined by polymorphisms, and inhibitory or activating profiles [41]. Each individual’s unique set of KIR receptors arises from the haplotypes they carry, resulting in a diverse repertoire of receptors tailored to recognize a broad range of pathogens.

The high diversity of KIR alleles is classified into two predominant haplotypes, namely, haplotype A (with five inhibitory receptors, two activating receptors, and one pseudogene) and haplotype B, which has greater diversity; both of these haplotypes are separated by the *KIR3DP1-KIR2DL4* intergenic region [42]. The presence of haplotypes A and B, and specifically the high diversity inside haplotype B, allow for maintaining a wide repertoire of KIR receptors. These two types of haplotypes have evolved through evolution, even in populations affected by bottlenecks and epidemic infections [43]. It may be assumed that activating KIRs protect the organism from infectious diseases and cancer in a simplistic manner but increase the probability of autoimmune diseases [44]. Table 1 illustrates the frequency range of each KIR gene and haplotype across different populations, including those from Europe, Africa, Asia, and Latin America. This comparative analysis highlights the variation in KIR gene frequencies among these populations, underscoring the importance of considering population-specific data when studying KIR diversity and its implications for health.

The prevalence of one or more haplotypes is the result of the selection of the most useful haplotype to provide protection against the most prevalent pathogens in each region and, consequently, in each population. Additionally, other mechanisms in which KIR/HLA interactions are involved, such as fertility and placentation, may play a relevant role in the evolution of one population, in which one or another haplotype predominates. Hence, evolution of pathogens and KIR haplotypes occurs in parallel, being clearly affected by epidemic outbreaks and by bottlenecks that cause genetic drift and facilitate some genetic combinations in reproduction against others. Therefore, pathogen-driven selection might be responsible for the current KIR worldwide distribution patterns [56]. Hence, it is necessary to know the characteristics of each population before applying the knowledge reached in other populations regarding a particular pathology. Concerning the response to microorganisms, as obviously not only a pathogen is present in a population, more than one haplotype may be overrepresented [57]. As a general approach, homozygosis of specific haplotypes is less favourable; therefore, heterozygosis is increased. In fact, heterozygosis is more common in populations under greater viral infection pressure [58].

### 3.3. KIR Ligands

HLA class I molecules act as ligands for many inhibitory KIRs (Figure 4). In regard to HLA-C molecules, two different ligand groups can be distinguished based on the amino acid situated at position 80. HLA-C1 ligands contain an asparagine at position 80 and are related to *KIR2DL2* and *KIR2DL3*, whereas HLA-C2 has a lysine residue and binds to *KIR2DL1* [59]. HLA-B molecules interact with the KIR receptor through the Bw4 epitope, binding *KIR3DL1* [60]. *HLA-A*03* and *HLA-A*11* have been reported to bind *KIR3DL2*, while *HLA-G* is responsible for the inhibitory signalling of *KIR2DL4* [61,62]. Allelic variation strongly influences ligand specificity. The binding of NK receptors to HLA class I molecules primarily induces an inhibitory response through self-recognition and generation of immune tolerance. However, HLA class I molecules can also interact with activating KIRs (Figure 4), increasing complexity.

## 4. HLA/KIR Interaction and Association with Disease

Because HLA genes are located on chromosome 6 and KIR genes are located on chromosome 19, these two groups of genes exhibit independent inheritance. The strength of the interaction between HLA and KIR genes is influenced by the presence/absence of KIR genes and allelic variability in both regions and plays a crucial role in susceptibility to disease. Additionally, this interaction may be affected by the specific peptides bound to the HLA molecule [44]. The relevance of the specific alleles involved in the development of different entities has been widely studied.

### 4.1. HLA/KIR and Reproduction

KIR/HLA-C binding has been described to be associated with preeclampsia and pregnancy failure, resulting in strong selection pressure [63,64]. In fact, *HLA-C* expression is low in all cells except for extravillous trophoblast (EVT) cells, where expression of *HLA-C* is high, and *HLA-A* and *HLA-B* are not expressed. EVT functions by transforming the arteries of the uterus into large vessels during placentation, a process affected by the interaction between the KIR genotype of the mother and the *HLA-C* haplotype present in the foetus [65]. An example of this is the association of the haplotype C1/C2 and specific KIR alleles with the ability of the trophoblast to invade the uterine mucosa to a successful pregnancy. Specifically, the maternal KIR genotype AA (absence or low number of activating KIR receptors) combined with a foetus carrying *HLA-C2* is associated with a greater risk of preeclampsia, foetal growth restriction, unexplained stillbirth, and recurrent miscarriage [63,65]. However, this effect is restricted to a foetus carrying almost one more *HLA-C2* allele than the mother. Thus, although the presence of *HLA-C2* is deleterious in the foetus, it is protective in the mother, promoting its persistence during evolution. For its part, *HLA-C1* preservation is promoted by its relevant role in the response to infection. In the same way, the deleterious effect of KIR haplotype A in successful pregnancy is countered by its protective role in infection [43]. With the exception of one study performed in the Uganda population that found that a group of C2-binding KIR2DS5 allotypes protected against pre-eclampsia compared to the non-binding KIR2DS5 allotypes [66]. Many studies on the influence of HLA-KIRs on reproduction have been conducted in European women [63,65,67,68,69,70,71]. While the effects of these factors have also been studied in other populations, such as African and Asian women [72,73,74,75], these studies are less extensive in comparison.

### 4.2. HLA/KIR and the Response to Infection

An example of the relevance of the KIR/HLA combined genotype in infection is the case of hepatitis virus C, where response to combined therapy (pegylated alpha interferon and ribavirin) is better in patients carrying the *KIR2DL3/KIR2DL3-HLA-C1C1* genotype and poorer in carriers of the *KIR2DL2/KIR2DL2-HLA-C1C2* genotype [76].

Another interesting case is tuberculosis, an inflammatory disease caused by *Mycobacterium tuberculosis* (Mtb), with a high prevalence worldwide [77]. A systematic review and meta-analysis of studies from Canada, Africa, Asia, and North America revealed associations between *KIR2DL3*, *KIR2DS1*, *KIR2DS4*, and *KIR3DL1* and increased risk of infection. These genes are in linkage disequilibrium inside haplotype A [78]. Most likely, a predominant inhibitory signal on NKs delays clearance of the mycobacterium and promotes progression of the disease. The African population has one of the highest worldwide tuberculosis incidences. Despite these findings, few studies on the role of HLA and KIRs have been conducted in underrepresented populations. In one of the few examples, in a study of Iranian patients, an association between inhibitory KIR expression and resistance to tuberculosis treatment was described [79].

In the case of HIV infection, a genotype including the *KIR3DL1* homozygous (hmz) genotype termed *KIR3DL1**h/*y cocarried with *HLA-B*57* (*h/*y + B*57) and the *KIR3DS1*hmz genotype has been associated with no infection in people with repeated contact with the virus [80]. Additionally, the presence of *KIR3DL1* and *HLA-Bw4* has been associated with slow progression of acquired immunodeficiency syndrome (AIDS) [81]. A study in HIV patients from South Africa showed a viral mechanism of evasion of the NK-mediated immune response by a mutation in the p24Gag viral peptide, which resulted in improved binding of the inhibitory NK cell receptor *KIR2DL3* to *HLA-C*03:04* and therefore inhibited the NK response [82]. In HIV-infected in Brazil, carriers of *HLA-C*07* but not *KIR2DL3* were found to exhibit active protection against tuberculosis onset [83].

### 4.3. HLA/KIR and Rejection Status

The role of KIR receptors in response to transplantation has been described as an association between the presence of the activating receptor *KIR2DS1* in the donor during haematopoietic stem-cell transplantation and a decreased rate of relapse in acute myeloid leukaemia, but only if the HLA-C1 antigen is present. No relevant graft-versus-leukaemia effect is detected when the donor is a carrier of the *HLA-C2/C2* genotype [84]. Similar results were described for KIR haplotype B by other authors [85,86]. However, this effect of graft-versus-leukaemia has not been detected in acute lymphoid leukaemia [87].

In the case of kidney transplantation, NKs are involved in antibody-mediated rejection since some of them express CD16, an FcgammaRIII receptor that interacts with anti-HLA preformed antibodies [88]. Additionally, NK cells are involved in acute T-cell-mediated rejection [89], with inhibitory KIR receptors playing a crucial role. It has been suggested that nonrecognition of HLA class I donor molecules by KIRs expressed by NK cells may induce NK cell alloreactivity against the graft; specifically, when the combinations of recipient *KIR2DL1* and donor *HLA-C2* and recipient *KIR3DL1* and donor HLA-Bw4 are not present, the risk of chronic rejection is increased due to a lack of NK inhibition [90]. Additionally, in a retrospective cohort study of 397 HLA-DR-compatible kidney transplantations, Van Bergen et al. described an increased risk of rejection when KIR-ligand mismatching occurred and did not permit suppression of NK-cell activity supported by inhibitory ligands [91].

### 4.4. HLA/KIR and Autoimmunity

Although NK cells have been described to be historically associated with the innate immune response, they are also involved in the increase and continuation of the adaptive immune response or in the generation of tolerance. In fact, expansion of pathogen-specific cells after the first encounter and generation of “memory” cells able to induce an increased secondary response upon contact with the antigen have been described [92]. Additionally, CD56^bright^ CD16^−^ cells have been proposed as “regulatory” NK cells [93], and a possible role for NK cells in the generation of autoimmunity has thus been proposed. Interestingly, in the case of autoimmune diseases, such as autoimmune hepatitis, the presence of the KIR2DS1-activating KIR gene and HLA-C2 ligands is significantly greater in patients than in controls [94].

In rheumatic diseases such as AS, epistatic interactions between KIR alleles and their HLA ligands have been described. A strong association between AS and the *HLA-B27* allele has been established worldwide, and this association between HLA molecules and disease is one of the best known associations [95]. However, these findings do not explain the totality of the genetic susceptibility found in AS or other HLA class I alleles, and other immune response-associated molecules associated with this disease have been described [96]. These other associations are especially relevant in populations in which the frequency of *HLA-B27* is low or nonexistent [24,97]. The association of *HLA-B27* with AS is modified by the presence of different KIR ligands, with overrepresentation of *KIR3DS1* in AS HLA-B27+ patients [98], and coexistence of the *HLA*-Bw4+ HLA-A*32 and *KIR3DL1* increases the risk of AS more than 3-fold [99]. In the Iranian population, several associations of KIR alleles and their HLA-C ligands with AS have also been described [100]. In PsA, an association between determined HLA-C and KIR alleles and PsA has been described, with the highest risk associated with the presence of the *KIR2DS2* allele and the absence of HLA-C ligands for homologous inhibitor KIRs [101].

### 4.5. HLA/KIR Diversity and Distribution Worldwide

As mentioned above, the majority of HLA/KIR and disease studies have focused predominantly on Caucasian and Asian populations. However, limited analyses have been conducted in Latin American and African cohorts, primarily addressing infectious diseases. Consequently, valuable genetic information is overlooked in these populations. Accordingly, there is a pressing need for additional research in African and Latin American populations.

The characteristics of the genome in Amerindian populations have evolved through genetic isolation from other populations and demographic events, such as founder effects and bottlenecks. Regarding HLA/KIR frequencies, the literature on Amerindian individuals is scarce, and we are far from achieving good comprehension of HLA/KIR diversity in these regions, which may contribute to elucidating the coevolution between HLA and KIR genes [102].

Variation within KIR genotypes was analysed a study encompassing six distinct South American native populations from Brazil and Paraguay, as well as two urban populations with European and Japanese ancestry in the southern region of Brazil, by means of custom next-generation sequencing method [103]. The findings revealed that many KIR alleles identified in the first six populations were absent in the urban samples. In terms of HLA/KIR interactions, the results show that Amerindians typically exhibit only 3 to 5 HLA/KIR interactions per individual. In contrast, urban Brazilian populations with European ancestry have an average of 6.9 HLA/KIR interactions, and those with Japanese ancestry have an average of 5.8. Among the HLA alleles involved, HLA-C contributes between 82.6% and 97.1% to the total population in the Amerindian population, whereas this contribution decreases to 51.3% and 57.8%, respectively, in urban populations [103].

In a mixed Colombian population of European (75%), African (7%), and Native American (18%) descent, characterization of KIRs based on polymerase chain reaction sequence-specific oligonucleotide (PCR-SSO) revealed *KIR2DP1* and *KIR2DL1* to be most prevalent, accounting for 77% of the population [55]. Consistent with the findings of previous studies, HLA-C was the most common ligand, along with HLA-Bw4, with an approximate frequency of 85%. The researchers also observed a low prevalence of KIR haplotype A, constituting only 13% of the population, aligning with findings in other Central and South American populations [55]. In another study involving a Brazilian cohort [45], researchers observed KIR frequencies exceeding 90% for *KIR2DL1*, *KIR2DL3*, *KIR3DL1* (inhibitory KIR), *KIR3DS1* (activating KIR), and *KIR2PD1* (pseudogene). These frequencies were compared with those in North, Central, and other South American populations, as well as Europe, Africa, Oceania, and Asia. The frequencies were largely similar for all KIR genes when compared to those in Caucasian populations. However, notable differences emerged when comparing these frequencies with those in Asian and African populations. For instance, the frequency of *KIR2DL2* in Brazil is approximately 60%, whereas it is approximately 20% in the Chinese population. However, it is also important to note that the use of different methodologies can influence the gene frequency values that are ultimately reported.

In studies in the Ugandan population [104], relevant differences were found in KIR frequencies when compared with those found in a UK cohort. Specifically, a lower frequency of *KIR2DS1* and *KIR3DS1*, both of which are associated with the KIR telomeric B region, was found in the Ugandan population. Studies in mixed population from South Africa (comparing populations with Caucasian, black African, Indian, and mixed ancestry) corroborate the lower frequency of *KIR2DS1* and *KIR3DS1* in black Africans than in the rest of the population [105]. These results are similar to those in other African populations [106,107,108]. *KIR3DS1*/*KIR3DL1* receptors are highly relevant in Africa due to their association with diseases such as tuberculosis and HIV infection [109].

## 5. Conclusions

The genetic variability within HLA and KIR regions, both integral components of the immune system, adds layers of complexity that recent high-powered GWASs have started to unravel owing to advances in SNP arrays and HLA reference panels. Nevertheless, our review highlights a significant gap in the existing knowledge. The majority of GWASs have disproportionately focused on populations of Caucasian origin, leaving a crucial lack in our understanding of the intricate genetic variations contributing to disease pathogenesis in underrepresented populations. This not only underscores existing disparities in health care accessibility but also poses challenges in identifying novel genetic variants relevant to disease aetiology within these neglected populations.

Given that the distinct KIR and HLA haplotypes prevalent in each population are shaped by specific environmental pressures, our review emphasizes the urgent need for increased research focused on the HLA/KIR axis in infection and autoimmune diseases, reproduction, and transplantation, particularly in underrepresented populations. Addressing this knowledge gap is not only a matter of scientific equity but also holds the key to unlocking new insights into disease mechanisms that might pave the way for more inclusive and effective health care strategies worldwide. The differences in disease symptoms or outcomes could be very high among racial groups. Thus, the inclusion of more ethnically diverse groups in clinical trials in order to represent as much population as possible will not only improve patient care, but also the efficiency of personalized medicine, an area within healthcare that is increasingly relevant.

## Figures and Tables

**Figure 1 biomedicines-12-01333-f001:**
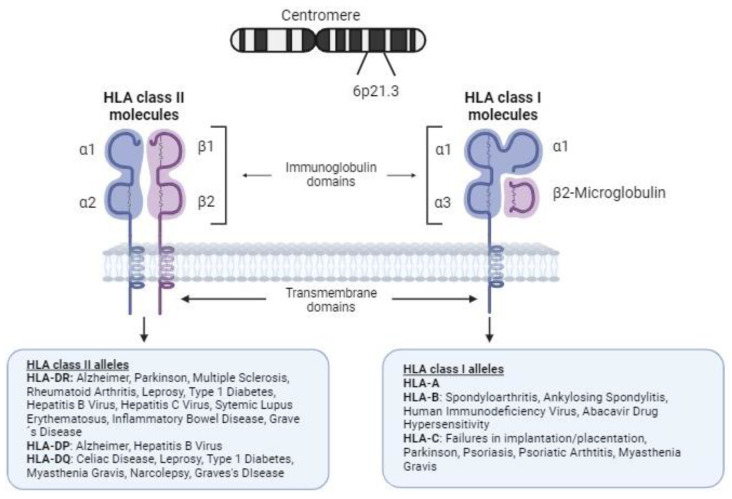
Structure of human leukocyte antigen class I and II molecules and association with autoimmune and infectious diseases. HLA class I molecules on the cell surface are composed of a heavy chain of 340 amino acids, the extracellular domains (immunoglobulin domains α1, α2, and α3) of which are encoded by exons 2, 3, and 4, respectively. Notably, there is a fourth domain provided by β2-microglobulin (encoded on chromosome 15), a light nonpolymorphic chain. The peptides resulting from degradation of the cell proteins are presented by these molecules to CD8 T cells (binding to the T-cell receptor, TCR) and NK cells (through their diverse membrane receptors), allowing them to tolerate self-antigens and promote a response to nonself antigens. HLA class II molecules are composed of two chains (an α-chain and a β-chain) with variable polymorphisms. HLA class II present processed peptides derived predominantly from extracellular and membrane proteins to CD4+ T cells.

**Figure 2 biomedicines-12-01333-f002:**
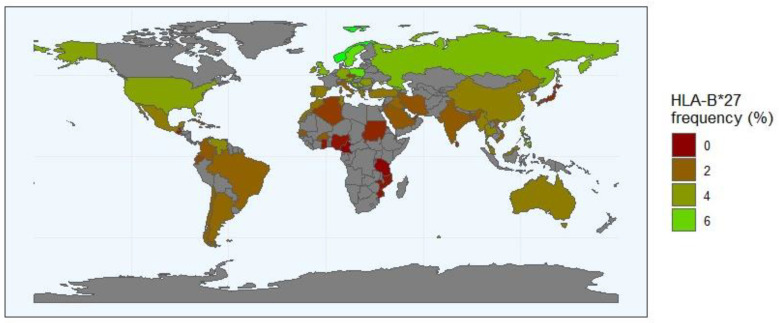
The frequency distribution of *HLA-B*27* allele worldwide (http://www.allelefrequencies.net/, accessed on 10 May 2024). The *HLA-B*27* frequency is represented in a scale from dark red (*B*27* allele rare or absent) to green (regions where the frequency of the *B*27* allele is the highest). The regions with a grey background indicate geographical locations where no data are reported. The numbers in the scale represent the frequency of the *HLA-*27* in the population.

**Figure 3 biomedicines-12-01333-f003:**
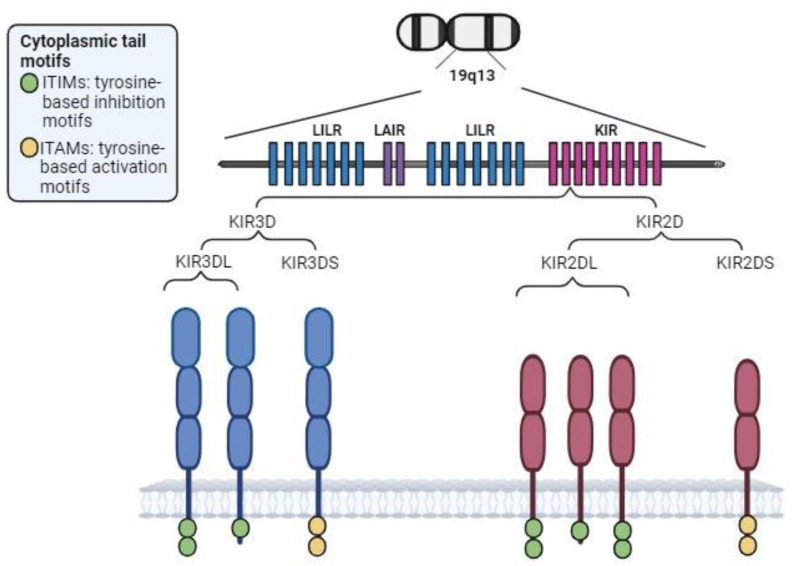
KIR region and structure. Depending on the longitude of its cytoplasmic tail, KIRs exhibit inhibitory (long cytoplasmic tail (L), including several tyrosine-based inhibition motifs (ITIMs)) or activating (short cytoplasmic tail (S), containing immunoreceptor tyrosine-based activation motifs (ITAMs)). KIR2DL4 is the exception: despite having a long cytoplasmic tail, it is an activating receptor. The ITAMs within activating KIRs are implicated in activation of the DAP-12 transduction chain, enhancing production of cytokines and chemokines.

**Figure 4 biomedicines-12-01333-f004:**
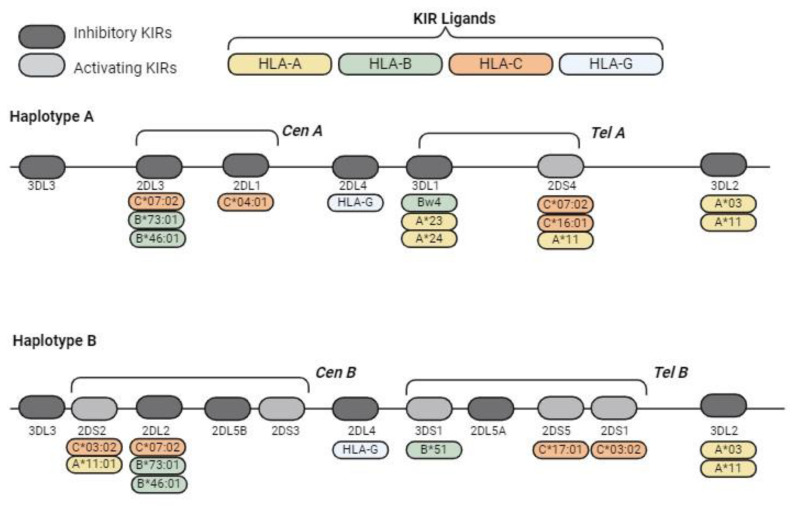
Schematic representations of the typical KIR haplotypes and KIR ligands. KIR genes exhibit a high recombination frequency between the centromeric region and the telomeric region because they are transposon-rich areas. These centromeric/telomeric regions and their haplotypes are classified into two predominant haplotypes, namely, haplotype A (with five inhibitory receptors, two activating receptors and one pseudogene) and haplotype B, which have greater diversity; both of these haplotypes were separated by the KIR3DP1-KIR2DL4 intergenic region.

**Table 1 biomedicines-12-01333-t001:** Frequency of KIR genes and genotypes in different populations.

**KIR Gene ***	**European** **Frequency (%)**	**African** **Frequency (%)**	**Asian** **Frequency (%)**	**Latin American** **Frequency (%)**	**References**
*KIR2DL1*	90–100	90–100	90–100	90–100	[45]
*KIR2DL2*	45–55	60–70	20–50	50–60	
*KIR2DL3*	85–95	80–90	80–100	80–90	
*KIR2DL4*	100	NT	100	100	
*KIR2DL5*	40–50	65–70	40–60	50–60	
*KIR2DS1*	30–50	20–30	30–50	30–50	
*KIR2DS2*	50–60	50–60	20–60	50–60	
*KIR2DS3*	20–40	20–30	10–40	25–35	
*KIR2DS4*	90–100	90–100	90–100	90–100	
*KIR2DS5*	20–40	50–60	20–40	30–40	
*KIR3DL1*	90–100	90–100	90–100	90–100	
*KIR3DL2*	100	NT	100	100	
*KIR3DL3*	100	NT	100	100	
*KIR3DS1*	30–50	10–20	30–40	30–50	
*KIR2DP1*	90–100	90–100	90–100	90–100	
*KIR3DP1*	100	NT	100	100	
**KIR Genotype ****	**European** **frequency (%)**	**African** **frequency (%)**	**Asian** **frequency (%)**	**Latin American** **frequency (%)**	
AA	35–55	55–75	55–65	40–65	[46,47,48,49,50,51,52,53,54,55]
Bx	45–65	25–50	35–45	35–60	

* European frequency was measured in Italian, French, English, and Croatian populations; African frequency was measured in the Ugandan population; Asian frequency was measured in Lebanese and Chinese populations; Latin American frequency was measured in Brazilian and Argentinian populations. ** European frequency was measured in Portuguese and Spanish populations; African frequency was measured in Sub-Saharan Ga-Adangbe populations and a Senegal cohort; Asian frequency is based on measurements in a cohort of Malaysians, Chinese, Indians, aboriginal individuals, and the Kuwaiti population; Latin American frequency was measured in Colombian and Brazilian populations. NT: not tested.

## Data Availability

No new data were created or analyzed in this study. Data sharing is not applicable to this article.

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
