# Peer review of "Unveiling the Significance of HLA and KIR Diversity in Underrepresented Populations"

_biomedicines, 2024, doi:10.3390/biomedicines12061333_

Round 1

Reviewer 1 Report

Comments and Suggestions for Authors

Review  on the  article "Unveiling the Significance of HLA and KIR Diversity in Underrepresented Populations":

The article is a review paper emphasizing the significance of comprehending the diversity of human leukocyte antigen (HLA) and killer-cell immunoglobulin-like receptor (KIR) genes among various ethnicities, “particularly focusing on underrepresented populations in Latin America”. The text explores the functions of HLA and KIR genes in the immune response, susceptibility to diseases, and the challenges faced by current genome-wide association studies (GWAS) in encompassing this variability. A lot of review articles have developed the aspects of the relationships between KIR and HLA ligands polymorphisms and autoimmune and viral infections. The authors did not explore the role of these two major immune complexes in cancer diseases.

 I agree with authors  that the underrepresentation of these populations in GWAS and HLA imputation studies results in disparities in healthcare access and hinders our comprehension of new genetic variations related to disease development.

The major issue that the review article, does not present original research findings or data.

Check for this sentence Page 2 “HLADP, DQ, and DR and are encoded by the gene pairs HLADRA and DRB, HLADQA and DQB, and HLADPA and DPB, respectively” As respectively is not respected

For the rs9277534 and rs4273729 please indicate the related genes

Page 8 “ll studies on the influence of HLA-KIRs on reproduction have been conducted in European women, and the effects of these factors in other populations are unknown”..

I completely disagree with authors, as in Africans, Asian and also American populations a lot of studies have been reported.

I recommend authors to add a comparative  table for KIR gene and haplotypes frequency among populations

I suggest reorganizing and summarizing specific portions, such as the background information on HLA and KIR structure and function. The parts discussing the intricate genetic arrangement of the HLA area, the extensive variety of known HLA and KIR alleles, and the technical explanations of HLA class I and II molecule structures might be shortened. This knowledge is widely known and extensively documented in numerous evaluations. Emphasize the crucial functions of HLA and KIR variation in the immune system and disease vulnerability, while condensing or relocating intricate technical information to additional materials.

This method would simplify the introduction and background, enabling the authors to allocate more space in the main text to the primary focus of the review - the importance of HLA and KIR diversity in marginalized populations and the constraints of current genetic research in representing this diversity. Condensing these background parts would enhance the manuscript's readability and coherence, therefore increasing its accessibility to a wider audience.

No information provided. There is a need to further explore the potential connections between HLA/KIR diversity and disease vulnerability, especially in under-researched communities.

No information provided. The conclusion part is concise and might be enhanced by offering more specific recommendations for future study paths.

Elaborate on the possible connections between HLA/KIR diversity and susceptibility to diseases, especially in marginalized communities.

Enhance the final statements by offering detailed suggestions for future research goals, partnerships, and technical improvements required to fill the knowledge gaps in this subject.

No information provided. Make sure to include all pertinent and up-to-date references, as the area is advancing quickly.

Comments on the Quality of English Language

Minor editing

Author Response

Comment 1. The major issue that the review article, does not present original research findings or data.

Reply 1. Thank you very much for your comment. As this manuscript is a review article, its primary objective is to synthesize and highlight the existing knowledge, with a specific focus on the disparities between well-studied populations (Asian and Caucasian) and underrepresented populations. We believe that emphasizing these gaps is crucial to guiding future research efforts and ensuring that these populations receive equitable attention in genetic studies. Our review aims to provide a comprehensive overview of the current state of knowledge and to underscore the importance of including diverse populations in GWAS and HLA imputation studies.

Comment 2. Check for this sentence Page 2 “HLA‑DP, ‑DQ, and ‑DR and are encoded by the gene pairs HLA‑DRA and DRB, HLA‑DQA and DQB, and HLA‑DPA and DPB, respectively” As respectively is not respected.

Reply 2. We apologize for the mistake. The sentence has been corrected in the text to: “HLA‑DR, ‑DQ, and ‑DP and are encoded by the gene pairs HLA‑DRA and DRB, HLA‑DQA and DQB, and HLA‑DPA and DPB, respectively”.

Comment 3. For the rs9277534 and rs4273729 please indicate the related genes

Reply 3. We have included the related genes for the SNPs rs9277534 and rs4273729 in the text. The revised sentences are as follows:

  • “… the rs9277534 SNP, located in the 3′ untranslated region of the HLA-DPB1 gene, is associated with …”
  • “… in which HLA-DRB1*03:01 in combination with rs4273729, located in an untranslated region of HLA class II genes, was shown to be …”

Comment 4. Page 8 “ll studies on the influence of HLA-KIRs on reproduction have been conducted in European women, and the effects of these factors in other populations are unknown”.. I completely disagree with authors, as in Africans, Asian and also American populations a lot of studies have been reported.

Reply 4. We apologize for the oversight. The sentence has been revised to reflect the broader scope of studies: “many studies on the influence of HLA-KIRs on reproduction have been conducted in European women, and while the effects of these factors in other populations have been studied, they are less extensively researched.”

Comment 5. I recommend authors to add a comparative table for KIR gene and haplotypes frequency among populations.

Reply 5. We appreciate the reviewer's suggestion. However, compiling a comprehensive comparative table for KIR gene and haplotypes frequencies among various populations would require an extensive amount of data from numerous studies and sources, which is beyond the scope of this review.

To facilitate readers who are interested in exploring KIR gene and haplotypes frequencies, we suggest referring to publicly available resources such as the Allele Frequencies Net Database (https://www.allelefrequencies.net/kir.asp). This resource provides comprehensive and up-to-date information on KIR frequencies across different populations. We have added a reference to this database in our manuscript to guide readers to this valuable resource for more detailed information.

However, if the reviewer still considers that an additional comparative table is necessary, we are willing to undertake the task and include it in the revised manuscript.

Comment 6. I suggest reorganizing and summarizing specific portions, such as the background information on HLA and KIR structure and function. The parts discussing the intricate genetic arrangement of the HLA area, the extensive variety of known HLA and KIR alleles, and the technical explanations of HLA class I and II molecule structures might be shortened. This knowledge is widely known and extensively documented in numerous evaluations. Emphasize the crucial functions of HLA and KIR variation in the immune system and disease vulnerability, while condensing or relocating intricate technical information to additional materials.

This method would simplify the introduction and background, enabling the authors to allocate more space in the main text to the primary focus of the review - the importance of HLA and KIR diversity in marginalized populations and the constraints of current genetic research in representing this diversity. Condensing these background parts would enhance the manuscript's readability and coherence, therefore increasing its accessibility to a wider audience. There is a need to further explore the potential connections between HLA/KIR diversity and disease vulnerability, especially in under-researched communities. The conclusion part is concise and might be enhanced by offering more specific recommendations for future study paths. Elaborate on the possible connections between HLA/KIR diversity and susceptibility to diseases, especially in marginalized communities.

Enhance the final statements by offering detailed suggestions for future research goals, partnerships, and technical improvements required to fill the knowledge gaps in this subject.

Reply 6. We appreciate the detailed suggestions provided by the reviewer. Following these recommendations, we have made several adjustments to improve the clarity and focus of the manuscript. Specifically:

  1. We have condensed sections on the structure and function of HLA and KIR molecules, reducing the technical details that are widely known and documented. This allows us to focus more on the critical roles of HLA and KIR variation in the immune system and disease susceptibility.
  2. We have emphasized the importance of HLA and KIR diversity in marginalized populations and the limitations of current genetic research in representing this diversity. This has been reflected in the introduction and main text to ensure these points are highlighted effectively.
  3. We have expanded the conclusion to include more specific recommendations for future research directions.

We believe these changes have improved the readability and coherence of the manuscript, making it more accessible to a wider audience. However, we are open to making additional revisions if the reviewer feels further adjustments are necessary.

Comment 7. Make sure to include all pertinent and up-to-date references, as the area is advancing quickly.

Reply 7. We have thoroughly revised the bibliography and added several up-to-date references to ensure that the manuscript reflects the most current research in the field.

Reviewer 2 Report

Comments and Suggestions for Authors

The study addresses a significant gap in the understanding of HLA and KIR diversity in underrepresented populations, a topic that has implications for personalized medicine, immunogenetics, and the management of diseases like autoimmune disorders and cancers.

This is a promising area of research that could lead to novel insights into the genetic underpinnings of disease susceptibility and treatment outcomes.

The manuscript "Unveiling the Significance of HLA and KIR Diversity in Underrepresented Populations" contains several areas that might need correction or improvement:

1.     The title  should not include line breaks between words (e.g., "Un-derrepresented").

2.     The manuscript could delve into how understanding HLA and KIR diversity can be directly applied to clinical practices, such as:

a.      Personalized Medicine: Discussion on how HLA and KIR genotyping could be integrated into personalized treatment plans, especially for autoimmune and infectious diseases.

b.     Population Genetics: Analysis of how migrations and genetic drift have shaped the current distributions and diversities of HLA and KIR alleles among different populations.

The manuscript by Santiago-Lamelas et al. is a valuable addition to the field of genetic research, particularly in its focus on underrepresented populations. With improvements in  methodological detail, the paper could significantly impact how genetic diversity is incorporated into clinical practices worldwide. It is recommended that the manuscript be accepted with minor revisions focused on expanding methodological details.

Author Response

Comment. The study addresses a significant gap in the understanding of HLA and KIR diversity in underrepresented populations, a topic that has implications for personalized medicine, immunogenetics, and the management of diseases like autoimmune disorders and cancers. This is a promising area of research that could lead to novel insights into the genetic underpinnings of disease susceptibility and treatment outcomes. The manuscript "Unveiling the Significance of HLA and KIR Diversity in Underrepresented Populations" contains several areas that might need correction or improvement:

  1. The title should not include line breaks between words (e.g., "Un-derrepresented").
  2. The manuscript could delve into how understanding HLA and KIR diversity can be directly applied to clinical practices, such as:
  3. Personalized Medicine: Discussion on how HLA and KIR genotyping could be integrated into personalized treatment plans, especially for autoimmune and infectious diseases.
  4. Population Genetics: Analysis of how migrations and genetic drift have shaped the current distributions and diversities of HLA and KIR alleles among different populations.

The manuscript by Santiago-Lamelas et al. is a valuable addition to the field of genetic research, particularly in its focus on underrepresented populations. With improvements in methodological detail, the paper could significantly impact how genetic diversity is incorporated into clinical practices worldwide. It is recommended that the manuscript be accepted with minor revisions focused on expanding methodological details.

Reply. We appreciate the reviewer’s insightful comments and suggestions. The issue with the title having line breaks (e.g., "Un-derrepresented") has been noted. We understand that this is a formatting issue likely related to the editorial process, and we will ensure it is corrected in the final submission.

Regarding the suggestion to delve into the application of HLA and KIR diversity in clinical practices: While the potential for HLA genotyping to be integrated into personalized treatment plans has been discussed in the manuscript, current evidence does not yet support the integration of KIR genotyping into clinical practice for autoimmune and infectious diseases. Therefore, we have not expanded this section to include speculative applications of KIR genotyping.

The manuscript also includes a discussion on genetic drift and its impact on the distribution and diversity of HLA and KIR alleles.

If the reviewer still feels that additional changes are necessary, we are willing to make further revisions as needed.

Reviewer 3 Report

Comments and Suggestions for Authors

The composition and frequency of HLA genotypes vary significantly among populations. Currently, studies on HLA/KIR and disease have predominantly focused on Caucasian and Asian populations. However, there is limited research investigating these issues in Latin American and African cohorts.

This review article raises the issue that HLA and KIR studies should also focus on these underrepresented populations.

Additionally, HLA-B*1502 has been reported to be associated with carbamazepine (CBZ)-induced Stevens-Johnson syndrome and toxic epidermal necrolysis (SJS/TEN) in the Asian population. Therefore, including this study in the review article is recommended.

Author Response

Comment. The composition and frequency of HLA genotypes vary significantly among populations. Currently, studies on HLA/KIR and disease have predominantly focused on Caucasian and Asian populations. However, there is limited research investigating these issues in Latin American and African cohorts. This review article raises the issue that HLA and KIR studies should also focus on these underrepresented populations. Additionally, HLA-B*1502 has been reported to be associated with carbamazepine (CBZ)-induced Stevens-Johnson syndrome and toxic epidermal necrolysis (SJS/TEN) in the Asian population. Therefore, including this study in the review article is recommended.

Reply. We appreciate the reviewer's suggestion. The article "Relationship between the HLA-B*1502 allele and carbamazepine-induced Stevens-Johnson syndrome and toxic epidermal necrolysis: a systematic review and meta-analysis" by Tangamornsuksan et al. has now been included in the manuscript.

Round 2

Reviewer 1 Report

Comments and Suggestions for Authors

The authors have tried to improve the first version, but it's not enough. 

This sentence is not sufficient: "Many studies on the influence of HLA-KIRs on reproduction have been conducted in European women, and while the effects of these factors in other populations have been studied, they are less extensively researched.". The authors did not add references for the studies they claimed.

in addition, the comparative table I recommended to authors for KIR gene and haplotypes frequency among selected populations was not added. There iis no need to refer to the "The Allele Frequency Net Database - Allele, haplotype and genotype frequencies in Worldwide Populations (allelefrequencies.net)"is not a suitable answer. Authors must select the populations that are comparable to underrepresented populations. 

 The authors should make these corrections before resubmitting the revised paper, otherwise, the paper will be rejected.

Comments on the Quality of English Language

Minor revision is requested

Author Response

Comment 1. This sentence is not sufficient: "Many studies on the influence of HLA-KIRs on reproduction have been conducted in European women, and while the effects of these factors in other populations have been studied, they are less extensively researched.". The authors did not add references for the studies they claimed.

Reply 1. We have revised the sentence regarding the studies on the influence of HLA-KIRs on reproduction to include specific references supporting our claims. The revised sentence now reads:

“Many studies on the influence of HLA-KIRs on reproduction have been conducted in European women [52,54,56–60]. While the effects of these factors have also been studied in other populations, such as African and Asian women [61–64], these studies are less extensive in comparison.”

Comment 2. In addition, the comparative table I recommended to authors for KIR gene and haplotypes frequency among selected populations was not added. There iis no need to refer to the "The Allele Frequency Net Database - Allele, haplotype and genotype frequencies in Worldwide Populations (allelefrequencies.net)"is not a suitable answer. Authors must select the populations that are comparable to underrepresented populations.

Reply 2. We have added a comparative table of KIR gene frequencies among selected populations. This table includes data from European, African, Asian, and Latin American populations. Thank you for your constructive suggestions, which have helped us improve the quality of our manuscript. We hope the revised version meets the reviewer's expectations.

Round 3

Reviewer 1 Report

Comments and Suggestions for Authors

The paper could be accepted

Comments on the Quality of English Language

No specific comments

Author Response

Thank you very much for considering our review for publishing it in the Biomedicines Journal.